# Carbon Nanotubes-Based Nanomaterials and Their Agricultural and Biotechnological Applications

**DOI:** 10.3390/ma13071679

**Published:** 2020-04-03

**Authors:** Dinesh K. Patel, Hye-Been Kim, Sayan Deb Dutta, Keya Ganguly, Ki-Taek Lim

**Affiliations:** Department of Biosystems Engineering, The Institute of Forest Science, College of Agriculture and Life Sciences, Kangwon National University, Chuncheon 24341, Korea; dineshbhud10@gmail.com (D.K.P.); asdl266@naver.com (H.-B.K.); sayan91dutta@gmail.com (S.D.D.); gkeya14@gmail.com (K.G.)

**Keywords:** carbon nanotubes, electrochemical, agricultural, biotechnological, engineering toolkits

## Abstract

Carbon nanotubes (CNTs) are considered a promising nanomaterial for diverse applications owing to their attractive physicochemical properties such as high surface area, superior mechanical and thermal strength, electrochemical activity, and so on. Different techniques like arc discharge, laser vaporization, chemical vapor deposition (CVD), and vapor phase growth are explored for the synthesis of CNTs. Each technique has advantages and disadvantages. The physicochemical properties of the synthesized CNTs are profoundly affected by the techniques used in the synthesis process. Here, we briefly described the standard methods applied in the synthesis of CNTs and their use in the agricultural and biotechnological fields. Notably, better seed germination or plant growth was noted in the presence of CNTs than the control. However, the exact mechanism of action is still unclear. Significant improvements in the electrochemical performances have been observed in CNTs-doped electrodes than those of pure. CNTs or their derivatives are also utilized in wastewater treatment. The high surface area and the presence of different functional groups in the functionalized CNTs facilitate the better adsorption of toxic metal ions or other chemical moieties. CNTs or their derivatives can be applied for the storage of hydrogen as an energy source. It has been observed that the temperature widely influences the hydrogen storage ability of CNTs. This review paper highlighted some recent development on electrochemical platforms over single-walled CNTs (SWCNTs), multi-walled CNTs (MWCNTs), and nanocomposites as a promising biomaterial in the field of agriculture and biotechnology. It is possible to tune the properties of carbon-based nanomaterials by functionalization of their structure to use as an engineering toolkit for different applications, including agricultural and biotechnological fields.

## 1. Introduction

Nanotechnology has an essential place in the progress of the latest technology, and is the leading investment field in all research fields. Nanotechnology provides an approach for inducing cell growth and forming a high-dimensional structure, like tissue engineering [1]. Among this nanotechnology, the prime spotlights are carbon nanotubes (CNTs) for industrial applications and implementations. CNTs are categorized into single-walled carbon nanotubes (SWCNTs) and multi-walled carbon nanotubes (MWCNTs) according to the number of layers present in the structure (Figure 1). SWCNTs consist of single-layer of graphene with the diameter range of 0.4–2 nm, whereas MWCNTs comprise a multilayer of graphene sheet with the outer and inner diameter of 2–100 nm and 1–3 nm, respectively, being 0.2 to several microns in length [2]. The physicochemical characteristics comparison of SWCNTs and MWCNTs is shown in Table 1. CNTs have been recognized as an attractive material that can be utilized in a variety of fields depending on mechanical [3], electrical [4], and thermal characteristics [5]. CNTs are well suited for biological applications, where a high aspect ratio is required [6]. Initially, CNTs’ works were primarily focused on electronic devices [7], displays [8], transistors [9,10], and so on, using the electrical characteristics of this nanomaterial. However, CNTs are considered a suitable material for several applications, ranging from biomedical [11] to agricultural technology [12,13,14]. The possible applications of CNTs are shown in Figure 2. Nanoparticles under 100 nm exhibit fascinating physical properties that open up new opportunities for application in various areas, including biological technology [15]. The development of new biocatalysts and drugs carrier from nanoparticles for bioengineering applications has advantages for grasping actions in the body, owing to their unique optical or electrical characteristics.

Various reports are available that emphasize the significance of CNTs to support bone growth by enhancing the mechanical properties of existing natural and synthetic polymers [16,17]. Currently, most of the studies on biological applications of CNTs-based and other carbon allotropes (graphene, fullerene) as biomaterials are focused on an approach to continuous interactions with living cells and tissues [17]. However, it has also been reported that cell/tissue interactions with CNTs can have adverse effects, which can cause a potential risk to human health [17]. Herein, we briefly described the synthesis and applications of carbon-derived nanomaterials in agricultural and biotechnological fields. The properties of CNTs are profoundly affected by their synthesis process. CNTs or their derivatives exhibited superior potential to promote the plant and growth. These nanomaterials can widely be explored in the fields of a nanosensor for the detection of the pathogens, as well as other bioresource fields, including battery, fuel cell, energy, and water purification.

## 2. Synthesis of CNTs

Various methods are available that are employed for the fabrication of CNTs from different carbon precursors. Each method has its advantages and disadvantages. Here, we briefly described some conventional methods utilized in the synthesis of CNTs with their merits and demerits.

### 2.1. Arc Discharge

This method is based on the potential difference between the two electrodes within a chamber. The graphite rod acts as an anode, and the migration of carbon particles migrated towards the cathode electrode, which is kept at a low temperature for the condensation of CNTs. The transition metals, such as Co, Ni, Fe, and Y, favor the formation of SWCNTs. The arc current sublimates the carbon precursor filled inside the anode and produces plasma at very high temperatures (~4000–6000 K) [18]. However, other by-products are also generated during the synthesis of CNTs through the arc discharge method. Therefore, it is essential to control the purification step for the synthesis of CNTs [19]. The high-quality CNTs can be produced through this approach by using the suitable catalyst and optimizing the process conditions [20,21]. The pressure of the gas and the applied current in the chamber are also important variables. As the pressure increases, the production of CNTs increases. However, a decrease in the yield of CNTs is observed at a very high pressure.

### 2.2. Laser Vaporization

In this method, the shooting of the targeted graphite is performed at about 1200 °C in a reaction furnace. As a consequence of this shooting, the vaporization of graphite has occurred, which is collected at a cold collector. Helium or argon gas is used for the carrier of the vaporized graphite, and the pressure of the reaction furnace is maintained at about 500 Torr [22]. Uniform SWCNTs can be produced in the presence of the transition metal catalysts such as Co, Ni, and Fe. The laser evaporation method is the optimal state for the high yield and precise control of process parameters [23,24]. The diameter of SWCNTs produced by the laser evaporation approach is profoundly affected by the furnace temperature and is generated in a narrow range with a diameter distribution of ~1.2–1.4 nm [25]. The diameter distribution of the produced materials can be easily tuned by changing the chemical compositions of the target, as well as the process gas [24].

### 2.3. Chemical Vapor Deposition (CVD)

A supportive catalyst is required for the synthesis of CNTs through the chemical vapor deposition (CVD) method from their carbon precursor. The decomposition of the injected gas accomplishes the synthesis of CNTs via heat and plasma [26,27]. The thermal CVD synthesis method is well-suited for the synthesis of highly pure materials, and the microstructures of the synthesized materials can be controlled in this method [28]. The temperature plays an essential role in the growth rate, diameter, and density of the developed CNTs [29]. An enhancement (~4 times) in the CNTs growth rate (0.5–2.0 μm/min) was noted by increasing the temperature from 750 °C to 950 °C. It has been noted that Ni has better catalytic activity than Co and Fe [30]. The thickness of the catalyst also has a significant influence on the density, diameter, and length of the developed CNTs [31]. It was noted that the thicker catalyst layer facilitates the formation of the larger diameter of CNTs with a shorter length. Further, the temperature gradients and catalyst–substrate interactions between catalyst particles are crucial for determining the CNTs’ growth mechanism [32]. The plasma CVD method has an advantage over the thermal CVD method [33], where a relatively lower temperature is required for the synthesis of CNTs [34].

### 2.4. Vapor Phase Growth

In the vapor phase growth process, the synthesis of CNTs takes place in the presence of the reaction gases and an organometallic catalyst in a reaction furnace without the assistance of any substrate. The graphite surface having CNTs is widely affected by the crystal face of the catalyst particle, whereas the diameter of the nanotubes is profoundly influenced by the size of the decomposed catalyst particles. This method has an advantage for the synthesis of CNTs [35].

The comparative study of these methods, including arc discharge, laser vaporization, chemical vapor deposition, and vapor phase growth, with their merits and demerits, are summarized in Table 2.

## 3. Properties

### 3.1. Electrical Properties

Several studies have determined the electrical properties of CNTs based on the concept of a helical structure, as proposed by Iijima [18]. The semiconductor or metallic potentials of CNTs are governed by the diameter and helicity of the graphene. As shown in Figure 3, CNTs can be made by the rolling of a graphene sheet such that the equivalent lattice parts of the two hexagons coincide [38,39]. The roll-up vector Ch = na1 + ma2 = (n, m) can control the diameter and helicity of the SWCNTs, where n and m are integers, and a1 and a2 are graphene lattice vectors [40]. The two integers (n and 0) correspond to the number of unit vectors along the direction of the grid [41]. The two (n, 0) exponents can be used to predict the electronic structure of SWCNTs. As shown in Figure 3, the chiral angle in the zigzag direction of the unit vector (a1, a2) of the hexagonal honeycomb grating is θ = 0° and the armchair tube corresponds to θ = 30° [42]. When (n, n), the nanotube is called “armchair,” and when (n, 0), the nanotube is called “zigzag” (armchair: conductor properties; zigzag: semiconductor properties). There are several reports available that show the high conductivity of CNTs [43,44,45]. It has been noted that the resistance of metal SWCNTs in rope form was about 10^−4^ Ω cm at 300 K. This value is a higher value than the current known conductive carbon fiber [46,47].

### 3.2. Thermal Properties

CNTs have better thermal conductivity than the diamond (sp^3^ hybridized) owing to the presence of sp^2^ hybridized covalent bonds [48,49]. The thermal conductivity of CNTs is widely influenced by the temperature and phonon mean-free path. The thermal conductivity value of SWCNTs is noted in the range of 1800–6000 W/m·K at room temperature. This value is higher than the diamond, 3320 W/m·K, which was known for the highest thermal conductive material. However, the thermal conductivity of MWCNTs is noted to be 3000 W/m·K [50,51]. The thermal properties of CNTs are also influenced by the functionalization [52]. The thermal conductivity of the polymer can be easily modified by incorporating CNTs in their matrix, and this potential is widely affected by the nature of CNTs [36,37,53].

### 3.3. Mechanical Properties

The strong covalent bond (sp^2^) enables the high mechanical strength of CNTs. It undergoes the bending condition without damaging its original structure after applying the strong force and returns the original condition as the force is removed from the surface. The average Young’s modulus values of CNTs with the diameter ranging from 1.0 to 1.5 nm were found to ~1.25 TPa, which is higher the in-plane modulus value of graphite [54,55]. The elastic properties of SWCNTs are overwhelmingly affected by the chirality and the diameter of CNTs [56,57]. The mechanical strength of CNTs varies with the size of the nanoparticles, and has a considerable impact on the mechanical strength of the composites [58,59]. Owing to the excellent characteristic, CNTs can be used not only as a reinforcing material, but also as an additive material.

## 4. Application of CNTs

### 4.1. Agriculture Applications

The unique properties of nanomaterials such as small size, large surface area, and reactivity provide excellent opportunities for its use in the agricultural sector. The foremost applications of CNTs in the agricultural field include seed germination, early plant growth, pesticides, and biosensor diagnostics and analysis. The potential toxicity of nanomaterials has not yet been widely investigated [60,61,62]. Here, we described the potential utilization of CNTs in the agricultural sector by considering some selected, but significant works.

#### 4.1.1. CNTs in Plant Growth

The applications of the nanomaterials as a promoter for plant and crop growth have received a significant amount of interest from the scientific community. It has been noted that CNTs can penetrate the thick seed coat and activate the water uptake process, which might be responsible for rapid seed germination and early growth [63]. Mondal and coworkers measured the seeds germination rate of *Brassica juncea* (mustard) in the presence of MWCNTs having a diameter of ~30 nm. A significant enhancement in the seeds germination rate, T_50_ (time for 50% germination), was noted in the presence of a low concentration of oxidized MWCNTs compared with the control. They observed that the moisture content was significantly high in oxidized MWCNTs-treated seeds than in the untreated condition, indicating that oxidized MWCNTs facilitated the water-absorbing potential of the seeds for rapid regeneration. The high water content in oxidized MWCNTs-treated seeds was the result of the easy penetration ability of these functionalized CNTs. However, the exact mechanism for the rapid growth of seeds in oxidized MWCNTs is still unclear. It is well known that aquaporins facilitate the water uptake inside the cells. The efficiency of aquaporin is profoundly affected by several factors like pH; concentrations of the heavy metal ions; osmotic pressure; and water channel expression genes such as plasma membrane intrinsic protein (PIP), small basic intrinsic protein (SIP), and so on. Aquaporin also reduces the flow of different ions through membranes and controls the electrochemical potential of the membrane. This potential of aquaporins is expected to be the key reason for the rapid regeneration of seeds in the presence of oxidized MWCNTs [64]. Several studies have been done to explore the effects of the various carbon nanomaterials (CNMs), including MWCNTs, fullerenes, and carbon nanohorns on different plants such as tomato, rice, cucumber, onion, radish, corn, soybean, switchgrass, and broccoli [65,66,67,68,69,70,71]. It was noted that 50–100 mg/L concentrations of CNMs are sufficient to penetrate the seeds for fast germination and growth rates [65,66]. Various factors such as size, shape, surface structure, solubility, and concentrations, as well as the presence of the functional groups, have significant contributions towards the toxicity and pathology caused by CNTs in the germination of seeds [61,72]. Functionalized carbon nanotubes (F-CNTs) also have an important aspect of being used as a nanomaterial to alter the seed germination and growth rates. Chang and coworkers have evaluated the toxic effects of CNTs (SWCNTs and MWCNTs) combined with cadmium (Cd) on wheat seedling growth. A significant reduction in total root length, root surface area, average root diameter, numbers of root hairs, and the dry weight of shoots and roots was observed in Cd-combined CNTs treatment groups than with Cd, as well as SWCNTs and MWCNTs treatment, indicating that Cd-combined CNTs remarkably inhibited wheat growth and development. Furthermore, a decrease in tubulins in the root was also noted. However, an enhancement in glutathione S-transferase and cytochrome P450 in the shoots and roots was observed in Cd-combined CNTs treatment groups, suggesting the improved defense ability of wheat seedling. It was interesting to see that the accumulation of Cd in shoot and root tissues was profoundly affected by the concentrations of CNTs. These results suggested that CNTs facilitated the toxicity of Cd to the wheat seedling. Therefore, the toxicity of CNTs should be remarkably considered with food security in the future with exposure of crops to Cd [73]. Transmission electron microscopy (TEM) morphologies of wheat plant cells under different conditions are shown in Figure 4a. The results indicated that CNTs had the potential to destroy the cell structure, and Cd highly influenced this ability. A comparative study has been done by Cano and coworkers to evaluate the effects of CNTs at various concentrations (0, 10, and 100 mg/kg) for the germination and growth of corn seeds. For this, they have taken pure SWCNTs, OH-functionalized, and surfactant stabilized SWCNTs [74]. The microwave-induced heating approach was explored to determine CNTs in different parts of the germinated seeds. They noted that the accumulation of F-CNTs in roots, stems, and leaves was independent of the functional groups present in CNTs, but dependent on the volume and composition of the soil. No significant difference in the plant physiological stress was observed between SWCNTs and the control. The effects of CNMs on plant and crop growth are also summarized in Table 3. Bioenergy crops are a suitable candidate for use in energy production. For bioenergy applications, plants should produce a high amount of biomass and resist adverse environmental conditions. The effects of CNMs on seed germination, biomass accumulation, and salt stress response of bioenergy crops (sorghum and switchgrass) were studied by Pandey et al. [75]. A significant enhancement in the germination rate was observed in CNTs-treated crops compared with the control, indicating the positive effect of nanomaterial towards crop growth. Approximately 73.68% and 31.57% enhancement in shoot biomass was noted in switchgrass seedlings with the exposure of CNTs for 10 days at concentrations of 50 and 200 μg/mL, respectively. A significant reduction in salt (NaCl)-induced stress symptoms was noted in CNMs-treated media compared with the control, demonstrating that CNMs have the potential to protect the plants against salt-induced stress in the saline growth medium. The effects of CNTs on the growth rate of switchgrass and sorghum seedlings at different concentrations after 10 days of exposure are given in Figure 4b,c.

#### 4.1.2. Biosensor

The biosensor is a device that quantitatively measures the molecules reacting in a solution having analytes to be measured by utilizing their reacting properties with a specific substance. The excellent physicochemical potentials make CNMs an ideal material for sensing applications to detect the pathogens [76,77]. In comparison with the commercially available sensors such as metal oxides, silicon, and so on, CNTs-based biosensors have significant advantages, such as high sensitivity (large surface area ratio), excellent luminescence properties, fast response time, and high stability [78]. Different types of sensors are explored for monitoring the pollutant/species present in the medium. Biosensors are utilized to detect compounds such as aromatic and organic compounds and halogenated pesticides. Solid-state electrochemical sensors are suitable for the chemical gas sensor from their sensitivity, reproducibility, and power consumption. The basic principle of a biosensor for soil diagnosis is to determine the relative activity of favorable and unfavorable microbe’s presence in the soil based on differential oxygen consumption owing to respiration. Surface plasmon resonance (SPR) phenomenon is also explored for the development of the biosensor from metallic nanoparticles [79]. Nano-biosensors are being rapidly explored in the agricultural sector and food processing. CNTs-based optical sensors were developed to monitor the real-time detection of pathogenic bacteria [80], organophosphate chemical warfare agents and pesticides [81], toxic materials, and proteins [82]. The one-dimensional (1D) properties of CNTs facilitate the ultrasensitive detection of analyte because all atoms are surface atoms, and minor perturbations in the chemical environment can dramatically change the electrical or optical properties [83]. This property plays a vital role in the monitoring of the optical sensor under various circumstances [84]. Among different biosensors, electrochemical biosensors are the most popular because of their excellent conductivity and electro-catalysis, high surface, and volume ratio [85]. The transfer of the electrons occurred in these biosensors [86,87,88]. CNMs have the potential to improve the response characteristics and can act as the immobilization matrices for the bio-receptors [89]. A significant decrease in the response time was observed in MWCNTs-coated electrodes used as a sensor [90]. An enhancement in the detection limits was noted in Au-MWCNTs nanocomposite, and it can detect concentrations up to 0.1 nM [91]. Enzymes are considered as a suitable substrate for the development of the biosensors. CNTs have been utilized as a support for the immobilization of enzymes in nanostructured devices. Scholl and coworkers have developed the thin film of CNTs for enhancing the enzymatic potential of penicillinase for biosensing applications. The presence of CNTs in the developed film not only altered the catalytic potential of penicillinase, but also facilitated their enzymatic activity. ConCap responses curves for penicillin G detection through the fabricated films are shown in Figure 5. Recently, Yang et al. [92] have reported a composite skin patch with a high-performance flexible sensor consisting of Ag/CNT/PDMS for monitoring of the heartbeat as well as breath during active labor (Figure 6). Owing to the presence of CNTs, the wrinkled patch is highly sensitive and conductive. This could potentially be used in prophylactic medicine for monitoring of fever or hyperthermia caused by specific pathogens. The biosensors developed with CNTs indicate regular steps of the distinct output signal for all concentration ranges compared with the control. These changes may directly influence the potential and performance of the developed sensor in terms of their sensitivity and coefficient of determination (R^2^) [93].

#### 4.1.3. Pesticide Analysis

The high adsorption properties of CNTs are utilized for extraction techniques such as solid-phase extraction (SPE) and solid-phase micro-extraction (SPME) [94]. SPE technology is one of the most widely used extraction methods for environmental, food, and biological sample pretreatment. Several studies have been done showing the potential of MWCNTs as a promising adsorbent for the pre-concentration of cobalt, nickel, and lead ions [95,96]; organophosphate (OP) pesticides [97]; and chloro-phenols [98]. The recoveries of the analyte were also altered by the amount of MWCNTs and the treatment conditions, indicating that, by varying the sample conditions, they could be extended to other analytes and other types of food samples [99]. SPE sorbent, based on nanoparticles, demonstrates the potential for adequate enrichment and sensitive analysis of metal ions in a variety of media [100,101]. The effects of the CNMs in the SPE technique are also given in Table 4. An enhancement in the extraction efficiency was noted in SWCNTs- or MWCNTs-coated SPME fiber. The development of fiber coating technology for high-efficiency extraction of the analyte is considered an exciting research direction in SPME [102]. Higher extraction efficiency, precision, and accuracy were observed in SWCNTs-coated fiber from the targeted samples [100]. It has been noted that CNTs-coated fibers have more extraction efficiency than the commercially available PDMS [103,104]. Saraji et al. synthesized CNTs/SiO_2_ nanohybrids for SPME coating and evaluated their extraction efficiency for some organophosphorus pesticides (OPPs) in vegetables, fruits, and water samples [105]. Gas chromatography-corona discharge ion mobility spectrometry was applied for the detections of the OPPs. Significant enhancement in the adsorption capacity and mass transfer rate was observed in CNTs/SiO_2_-coated SPME compared with the commercial SPME fibers (PA, PDMS, and PDMS-DVB), indicating their improved extraction efficiency. For water samples, the detection limits range was 0.005–0.020 μg/L, and the quantification limits were 0.010 and 0.050 μg/L, with excellent linearity in the range of 0.01–3.0 μg/L for the samples. The spiking recoveries range was from 79 (±9) to 99 (±8). Therefore, the developed materials have the potential and can be applied for the analysis of OPPs in real samples [106]. The influence of the CNMs in the SPME technique is also summarized in Table 5. Feria and colleagues have determined the presence of different types of pesticides in virgin olive oils using MWCNTs and carboxylated c-SWCNTs. It was interesting to note that the c-SWCNTs exhibited better sorbent capabilities than those of MWCNTs owing to the presence of carboxyl functional groups in their structure, which facilitates better interactions between pesticides and CNTs. A comparison of the performance of c-SWCNTs and MWCNTs for the detection of different pesticides from virgin oil samples is shown in Figure 7a. The bar diagram demonstrates the better sorbent potential of c-SWCNTs than MWCNTs for different kinds of pesticides from the selected samples owing to the presence of the different functional groups. The effect of the number of c-SWCNTs (10 and 50 mg) on the analytical signal for different pesticides is shown in Figure 7b. An enhancement in the peak area was observed by increasing the number of c-SWCNTs for all analytes up to 30 mg. Furthermore, a decrease in the peak value was noted after a 30 mg dose of c-SWCNTs owing to non-quantitative elution of the retained analytes [107].

### 4.2. Energy and Environmental Applications

Works on CNTs in the field of bioresources are being studied as a material capable of overcoming the limitations of existing carbon materials or improving performance by using the high electrical conductivity of CNTs. As CNTs showed a high specific surface area, much research has been conducted into CNTs as an adsorbent for the removal of different contaminants such as Zn^2+^ and Pb^2+^ [108]. CNTs nanocomposites have a wide range of applications depending on the type and combination of the target materials. Here, we have briefly described the nanotechnological applications of CNMs-based materials, including the battery, wastewater treatment, fuel cell, and energy storage, by considering some attractive works.

#### 4.2.1. Battery

Despite the rapid development of lithium-ion batteries, which have high power and energy density properties [109], numerous reports have focused on the application of CNTs for the energy sector [110,111,112]. The energy efficiency of CNTs is intensely affected by the synthesis method, shape, and structure. Maurin et al. showed that lithium was intercalated between the graphene layers of the MWCNTs prepared by arc discharge using micro-raman spectroscopy [113]. CNTs produced by the arc discharge method had a reversible capacity of 125 mA·hg^−1^ at a low current density [114], which has limited the practical application in lithium-ion batteries to some extent [115]. However, CNTs synthesized by chemical vapor deposition (CVD) showed the high reversible capacity of 340–640 mA·hg^−1^ at a low current density [116,117,118]. A comparative study was performed by Yang et al. using short CNTs (S-CNTs) and long CNTs (L-CNTs) synthesized through co-pyrolysis, as well as the CVD method, respectively, to evaluate the reversible capacity of both samples at a low current density. The reversible capacity of S-CNTs anode material was 266 and 170 mA·hg^−1^ at the current density of 0.2 and 0.8 mA·cm^−2^, respectively, which were twice that of L-CNTs anode materials. The surface film and charge-transfer resistant of S-CNTs anode materials were 1.7 Ω and 3–4 Ω, respectively, which is much lower than the L-CNTs (14 Ω, and 31.2–61.2 Ω) anode materials, indicating higher electrochemical activity [119]. The holes in the graphene sheet allow lithium to diffuse better inside the CNTs and increase the capacity. The conductive SWCNTs were able to store about five times more lithium ions than semiconducting SWCNTs [120]. The high conductivity of CNTs also provides enhanced electron transfer with nanostructured anode material [121]. However, long-term stability has remained a challenging task. The electrochemical performance is highly dependent on the nanostructure, shape, and surface properties [122,123,124,125,126]. Lee et al. have developed CNT–Si composite anode with extremely stable long-term cycling and a discharge capacity of 2364 mA·hg^−1^ at a tap density of 1.103 g cm^−3^. The CNT–Si composite anode retained an excellent cyclic maintenance equivalent to 90% of the initial discharge capacity after 100 cycles. A two-sloped full concentration gradient (TSFCG), Li[Ni_0.85_Co_0.05_Mn_0.10_] O_2_ cathode, was used to prepare the fuel cell configuration. The assembled fuel cell exhibited an energy density of 350 W h kg^−1^ with excellent capacity retention for 500 cycles at 1C [127]. The electrochemical performances of CNTs-based Li-ion batteries are given in Table 6.

#### 4.2.2. Wastewater Treatment

Nanotechnology plays a vital role in water purification. CNTs can be used for the purification of wastewater [128]. Adsorption and degradation/detoxification is the key strategy for the removal of contaminant from the samples through CNTs. The functionalization of the material can improve the efficiency of CNTs for contaminants. It is possible to target a specific contaminant through the well-modified CNTs. A schematic representation of CNTs’ modifications for the removal of contaminant from water and wastewater is shown in Figure 8. Design or modification of CNTs’ properties may also assist in the separation of materials following the contaminant treatment process. Nanoparticle separation is facilitated by incorporating a magnetic component into CNTs [129]. It is easy to control the potential and current in the electrochemical technique for wastewater treatment [130,131]. Yang et al. have used a seepage carbon nanotube electrode (SCNE) reactor to improve the electrochemical wastewater treatment efficiency. The innovative concept behind the reactor design was that the overall mass transfer would be significantly improved via contaminant migration through the porous carbon nanotube electrode. The current efficiency of the SCNE reactor was 340–519% higher than those of the conventional reactor, and the energy utilization to mineralize the equal weight of organic content was only 16.5–22.3% of the conventional reactor. The developed reactor has the potential for application in wastewater treatment [132]. The electrocoagulation is also useful for removing effluents from the polluted water [133]. These applications utilize the advantages of CNTs’ properties such as high reactivity, strong adsorption, and high specific surface area [133,134,135]. Zhang et al. have fabricated Ti/SnO_2_-Sb-CNTs electrodes for anodic oxidation of dye-containing wastewater through the pulse electrodeposition method. The CNTs-modified electrode exhibited a larger surface area compared with that without CNTs, which provides a more active area for electrochemical oxidation of organic pollutants. The CNTs-modified electrode was 4.8 times more durable compared with that without CNTs. The modified electrode has a higher kinetic rate constant, chemical oxygen demand (COD), total organic carbon (TOC) removals, and current mineralization efficiency, which are 1.93, 1.27, 1.26, and 1.38 times higher, respectively, than those of the unmodified electrode. The CNTs-based electrode exhibited 1.15 times more permeation flux compared with the electrode without CNTs [136]. The electrochemically activated CNTs filters were developed for wastewater treatment [137]. Thus, the solutions for implementing water reuse, seawater desalination, and water purification more efficiently and cost-effectively are expected to emerge from the use of nanotechnology with CNTs. The applications of CNMs in wastewater treatment are also summarized in Table 7. It was noted that phenolic compounds are often explored in the commercial manufacturing of several products such as resins, polymeric materials, ion exchange resin, dyes, drugs, and explosives, among others. Owing to the extensive uses of phenolic products, a large amount of phenol is discharged from industries in the water, which causes toxicity and can damage the cellular proteins. Therefore, the removal of phenolic compounds from the contaminated water on a large scale is necessary for a healthy life. For this, CNTs with rich pore structure, analytic abilities, high surface area, and sharp curvatures show great potential for the removal of the phenolic compounds from the contaminated water through π–π, electrostatic, hydrophobic, and hydrogen bonding interactions [138]. Ma et al. have prepared CNTs/Fe@C hybrids material for the removal of the binary dye from the contaminated water through the one-pot method with a high specific surface area (186.3 m^2^/g). A significant difference between single and binary dye systems was noted through the adsorption technique. The primary adsorption potentials of the prepared hybrids for the methylene blue (MB), methyl orange (MO), and neutral red (NR) were 132.58, 16.53, and 98.81 mg/g, respectively, and the adsorption equilibrium times were 80, 40, and 10 min, respectively. The adsorption capacity and their changes in single and binary dye systems are given in Figure 9a. Cooperative adsorption was noted in the MB–MO dye system through the developed hybrids material. An enhancement in the adsorption capacity was observed in the MB–MO dye system by 30% and 35%, with a decrease in the equilibrium time by 25% and 50%. Meanwhile, the MB–NR dye system exhibited a competitive adsorption tendency. The adsorption isotherm of MO and MB from the prepared hybrids material is shown in Figure 9b. These results suggested that the prepared hybrids had the efficiency to be used as a promising adsorbent for the large-scale applications in binary dye systems, which exhibited a cooperative and competitive adsorption tendency to address the dye pollution effectively [139]. Lee and coworkers fabricated MWCNTs-based polyaniline (PANi)/polyether sulfone (PES) membranes by in situ polymerization of aniline in the presence of MWCNTs for the effective removal of natural organic matter (NOM) in water. The fabricated membranes exhibited 30 times greater efficiency than the PES membrane. This enhancement was attributed to the synergistic effects of the MWCNTs/PANi complex. The electrostatic interactions between the membrane surface and NOM facilitate the adsorption capacity of the developed membrane. The fabricated membrane exhibited 100% water flux recovery and 65% total fouling ratio after treatment with 0.1 M HCl/0.1 M NaOH solution for 1 h [140]. The extending exploration of SWCNTs raises environmental concerns. Qu et al. have evaluated the microbial communities’ *(Zoogloea*, *Rudaea*, *Mobilicocus*, *Burkholderia, Singulisphaera*, *Labrys*, and *Mucilaginibacter)* responses of SWCNTs in phenol containing wastewater media. The enhancement in the phenol removal rates was observed in the SWCNTs-treated batch in 20 days initially. However, as the phenol concentrations increased to 1000 mg/L after 60 days, a decrease in the phenol removal rate was noted even at the higher concentration of SWCNTs (3.5 g/L). It was noted that SWCNTs protected the microbes from inactivation by generating more bound extracellular polymeric substances (EPSs), which form a protective layer for the microbes. A significant decrease in the bacterial community structure was observed after the addition of SWCNTs. This phenomenon is associated with the change in sludge settling, aromatic degradation, and EPS generation. These results demonstrated that SWCNTs exhibited the protective response for sludge microbes in phenol containing wastewater media and enabled the important information related to the potential effects of SWCNTs on wastewater treatment processes [141].

#### 4.2.3. Microbial Fuel Cells (MFCs)

Microbial fuel cell (MFC) technology produces hydrogen or electrons by a bacterial oxidizing process from substances such as wastewater. This is the basic concept of generating electricity through an anode–cathode system. For this, the cathode should have excellent compatibility with microorganisms and possess a large specific surface area per unit volume, as well as excellent durability as chemically safety materials [142]. CNTs have received much attention for cathodic applications owing to their superior and tunable physiochemical potential. The electronic signal is also affected by the temperature of the medium [143]. The high limiting current density and electrochemical performance were observed in the deformed CNTs owing to the higher specific surface area generated by deformation [144]. The modification is required in CNMs to achieve the proper catalytic surface area for better electrochemical performance [145,146,147]. The change in the aspect ratio and surface area of CNTs was performed using a metal catalyst such as platinum (Pt) [148,149,150]. The CNTs/Pt composites exhibited better powder density (~8.7% higher) than the pure Pt catalyst when the chemical oxygen content of the substrate reached 100 mg/L. The significant enhancement in the electrical properties was observed in nitrogen-doped CNTs [151]. The nitrogen-doped CNTs exhibited a maximum power density of 1600 ± 50 mW·m^−2^, which is significantly higher than the commonly used Pt catalyst for cathode application. For a better reaction process, the surface area and durability of the anodic materials should be high [152,153]. The CNTs/polyaniline (PANi) composites showed an enhanced electrochemical activity at a higher content of CNTs in the medium [154]. The CNTs-coated anode demonstrated ~62% higher voltage output than the untreated anode [155]. The performance of the anodic materials can also be improved by using the three-dimensional (3D) structure of graphene oxide (GO)/CNTs and melamine sponge composites [156]. The 3D graphene oxide (GO)/CNTs and melamine sponge display the highest electrochemical performance at a thickness of 1.5 mm. The porous structure facilitates the biocompatibility of the composites. These results provide valuable insights into the active anode–cathode development for MFC applications. The effects of the carbon-based electrode on MFCs are also given in Table 8.

#### 4.2.4. High-Efficiency Electrical Devices

For energy applications, it is crucial to increase the energy density of the material without compromising other electrochemical properties [157]. CNTs are not only light in weight, but also have a sufficient area for hydrogen storing in their tubular structure, which can increase the charge storage capacity per unit mass [158,159,160,161,162,163,164,165,166]. CNTs can also be utilized in other electrochemical applications [167,168] and supercapacitor preparation [169,170,171,172]. An increased surface area of CNMs is required for energy applications with pore sizes of 0.7 to 0.9 nm, which are suitable for the ions approach. It has been proved that hydrogen is stored in the pores formed in the space between the tubes, and the adsorbed hydrogen molecules are subjected to a stable surface suction force. Approximately 3.3 wt.% and 0.7 wt.% hydrogen adsorption was noted within the tube (10, 10) and interstitial space of CNTs, respectively [160]. A hierarchical structure is required to obtain the high output characteristics, which are connected in a vast pore region for the fast ion diffusion even at a high current density. The maximum power density can also be improved by using the cetrimonium bromide (CTAB) with CNTs [173]. A porous three-dimensional structure was formed by intercalating the CNTs into graphite in a vertical direction to improve the maximum energy density. A significant enhancement in the maximum energy density was observed in this structure, which was 117.2 Wh/L at a maximum power density of 424 kW/L per volume, and a maximum energy density of 110.6 Wh/kg at a maximum power density of 400 kW/kg per weight. This kind of structure is light in weight, which provides additional advantages to make small portable electronic products such as automobile batteries, rechargeable batteries, and notebook computers. The hydrogen storage capacity of different types of CNTs is given in Table 9. The hydrogen storage ability of CNTs is shown in Figure 10. The interaction energy plays a vital role in the storage of hydrogen. The results indicated that CNTs could effectively store hydrogen under cryogenic conditions, which is not suitable for mobile applications. This is because of the reduced interaction energy (1 kcal/mol) between hydrogen and the CNTs. For significant, but reversible storage under ambient conditions, the interaction energies should be around 7 kcal/mol. The interaction energy can be tuned by doping with heteroatoms or by incorporating light metal ions in CNTs [174].

## 5. Conclusions

CNTs have received a significant amount of interest in various applications owing to their superior physiochemical properties. Notably, the physicochemical properties of CNTs are profoundly affected by the diameter and helicity of the graphene sheet, as well as the number of graphene layers. The significant enhancement in the seeds germination/plant growth was noted in the presence of carbon-based nanomaterials compared with the control owing to the penetration of the seed coat, which allows more water uptake. However, the exact mechanism of action is still unclear. The CNTs-based sensor exhibited high sensitivity and stability, fast response time, and excellent luminescence properties. The high adsorption potential of CNTs facilitates the extraction process and is widely explored in the extraction technique for the removal of contaminants from the samples. CNTs or their derivatives are often utilized in the nanotechnology sector to develop high-efficient battery, fuel cells, electrode reactor for wastewater treatment, and energy storage. Notably, better electrochemical performances were observed in CNTs-based electrode compared with the control. CNTs can store hydrogen molecules in their structure, and this potential can be tuned by changing the electronic environment of CNTs.

## Figures and Tables

**Figure 1 materials-13-01679-f001:**
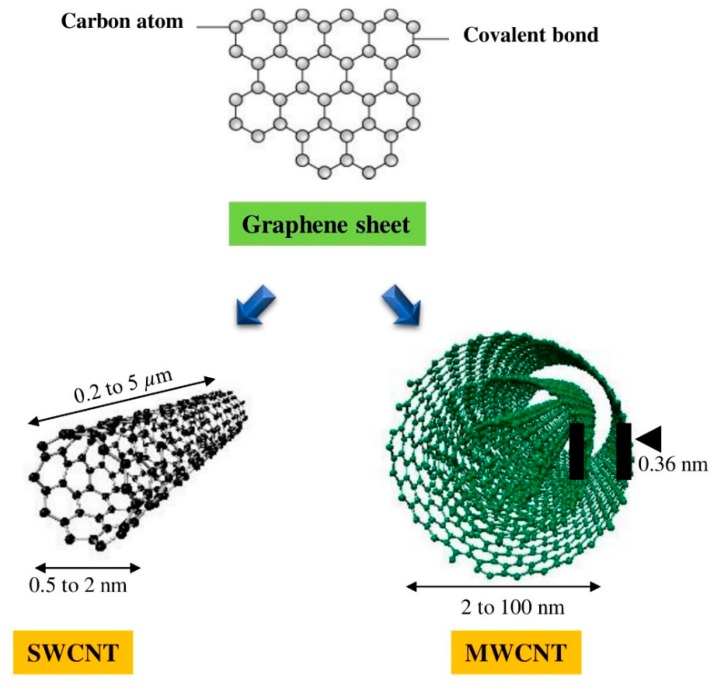
The conceptual diagram showing the general dimensions of the length and width of single-walled carbon nanotubes (SWCNTs) and multi-walled CNTs (MWCNTs) [2].

**Figure 2 materials-13-01679-f002:**
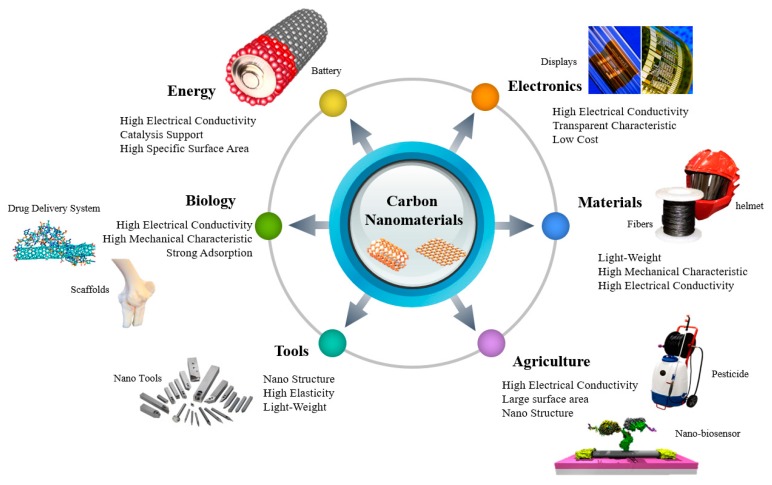
An overview of the properties of CNTs and with synthetic and transdermal applications. Various properties of CNTs enabling them to be used as the transdermal applications are depicted. Additionally, synthetic applications of CNTs are also depicted.

**Figure 3 materials-13-01679-f003:**
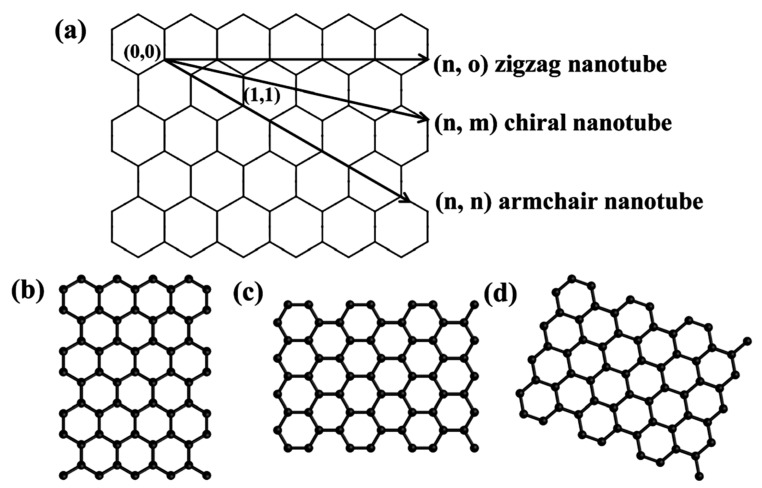
(**a**) Schematic representation of how a graphene sheet is rolled to form three chiralities of nanotubes: (**b**) zigzag, (**c**) armchair, and (**d**) chiral nanotubes [39].

**Figure 4 materials-13-01679-f004:**
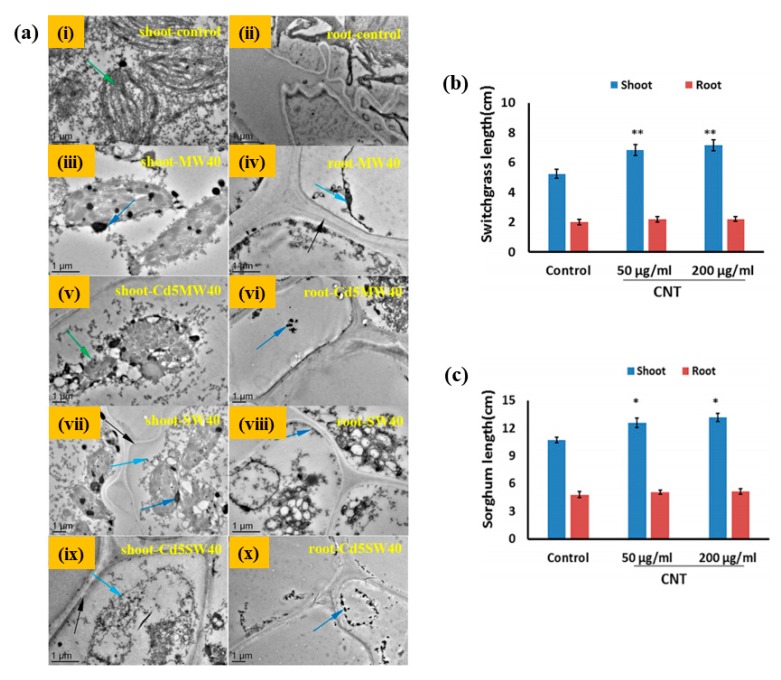
(**a**) Transmission electron microscopy images of plant cells. Green arrows indicate chloroplasts in (**i**,**v**); black arrows indicate the cell wall for (**iv**,**vii**,**ix**); navy blue arrows indicate CNTs deposition for (**iii**,**vi**,**vii**,**viii**,**x**); and light blue arrows indicate cell membrane in (**iv**,**vii**,**ix**) [73]. (**ii**) Growth enhancement on (**b**) switch grass and (**c**) sorghum seedlings by exposure to carbon-based nanomaterials. Effects on growth of bioenergy crops by CNTs added to growth medium. Measurements were performed on 10-day-old seedlings (n = 30 for both sorghum and switch grass). (* *p* < 0.05 and ** *p* < 0.01) [75].

**Figure 5 materials-13-01679-f005:**
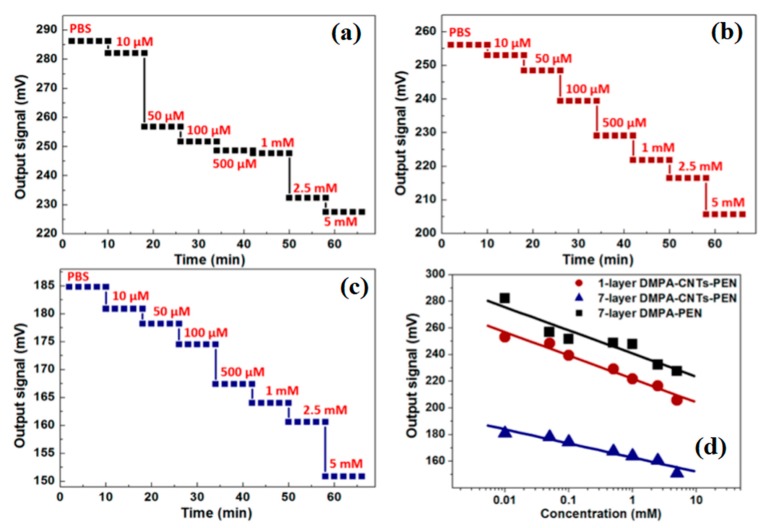
ConCap response curves toward penicillin G detection at different concentrations for LB film-incorporated EIS sensors: (**a**) seven-layer DMPA-penicillinase; (**b**) one-layer DMPA-CNTs-PEN; and (**c**) seven-layer DMPA-CNTs-PEN. (**d**) Calibration curve of ConCap responses for the LB films correlated with penicillin concentrations [93].

**Figure 6 materials-13-01679-f006:**
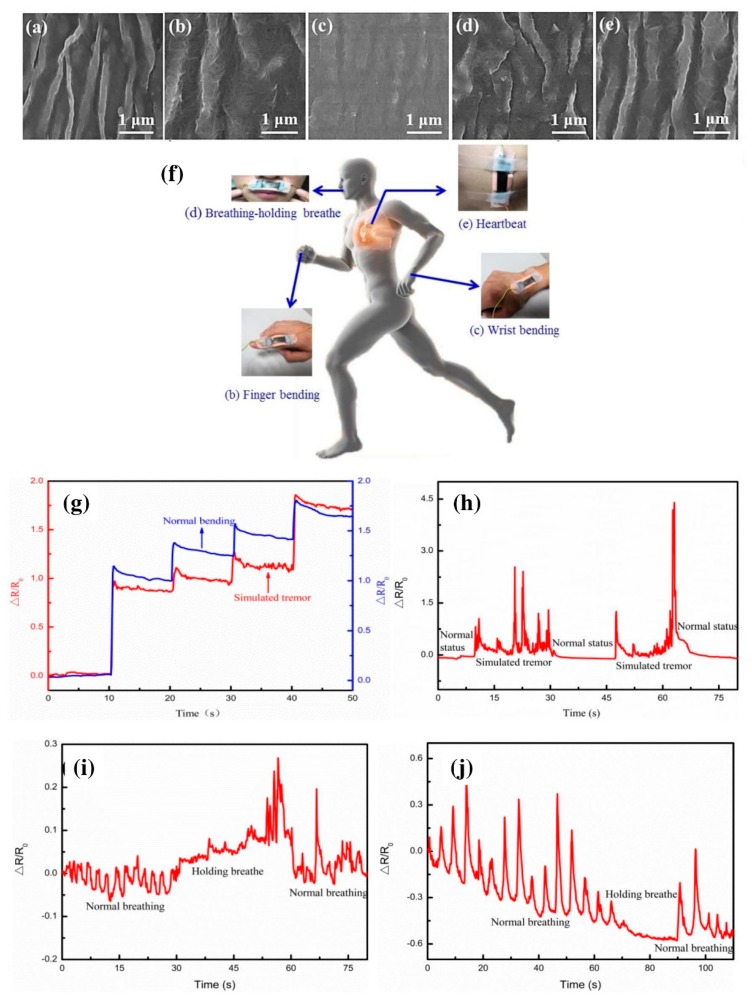
Ag-CNT-PDMS-based wearable sensors for monitoring the physiological conditions of the human body. (**a**–**e**) SEM morphologies of the wrinkled CNTs (left) and Ag/CNT/PDMS nanocomposite films under variant strain and release conditions (1 µm). (**f**) Schematic illustration of sensor location and its application. (**g**) Signals received from the finger-bending involving the normal bending (blue) and stimulated bending (red). (**h–j**) Signal from wrist, upper lip, and chest showing the significant change in peak [92].

**Figure 7 materials-13-01679-f007:**
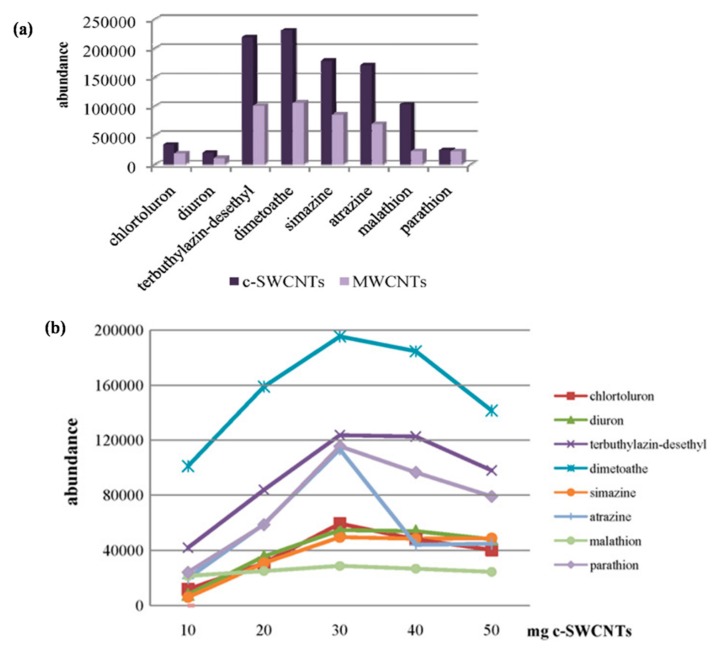
(**a**) Comparison of the performance of carboxylated (c)-SWCNTs and MWCNTs for the isolation of the selected pesticides from virgin olive oil samples. (**b**) Influence of the amount of c-SWCNTs packed in the solid-phase extraction (SPE) cartridge for the preconcentration of the selected pesticides from virgin olive oil samples [107].

**Figure 8 materials-13-01679-f008:**
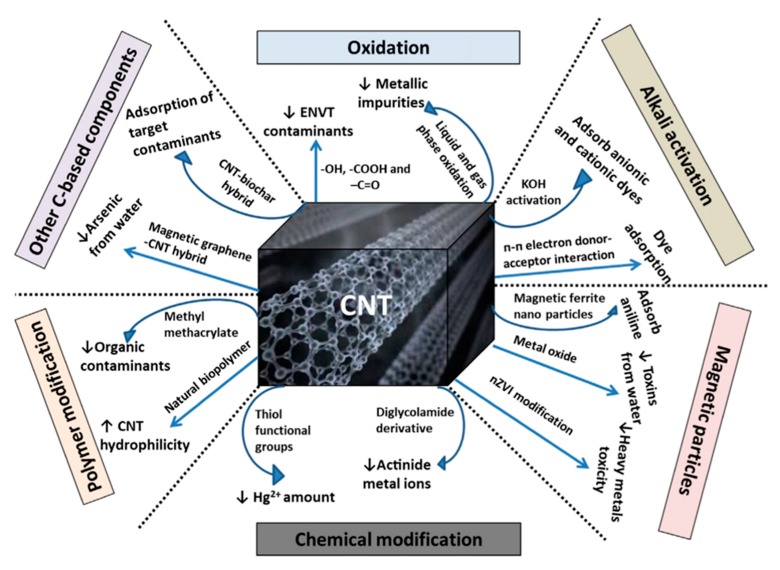
Schematic diagram representing different modification processes of CNTs for contaminant removal from water and wastewater (C: carbon; CNT: carbon nanotube; ENVT: environmental; Hg: mercury; KOH: potassium hydroxide) [129].

**Figure 9 materials-13-01679-f009:**
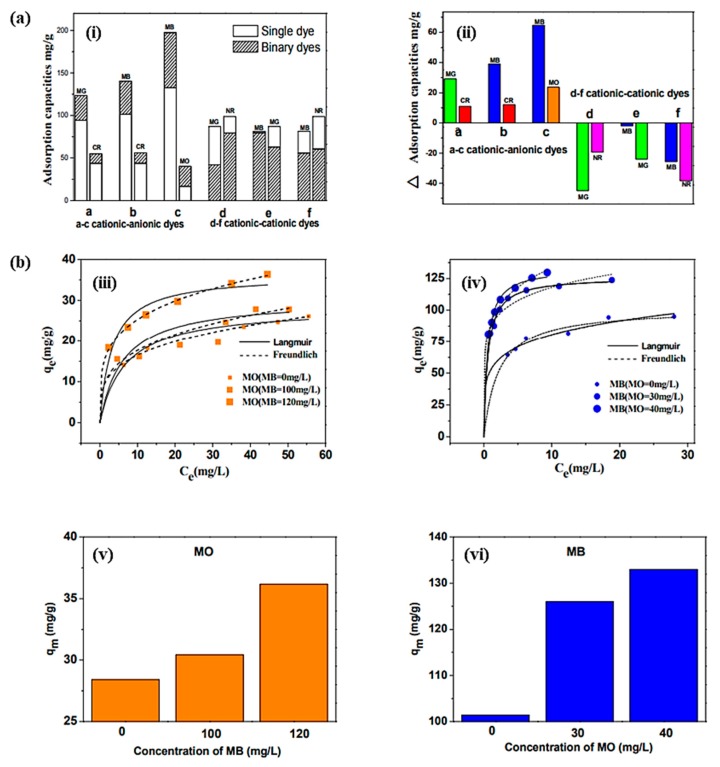
(**a**) (**i**) Adsorption capacities and (**ii**) their changes in single and binary dye systems. (**b**) Adsorption isotherms of methyl orange (MO) (**iii**) and methylene blue (MB) (**iv**) on CNTs/Fe@C fitted by the Langmuir and Freundlich models. The maximum adsorption capacity of MO (**v**) and MB (**vi**) increased with the concentration of the other dye [139].

**Figure 10 materials-13-01679-f010:**
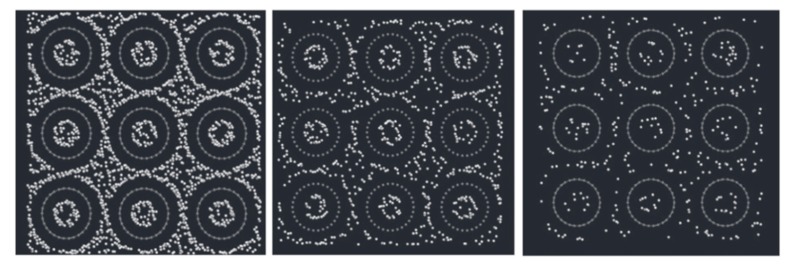
Hydrogen storage in nanotube bundles. Snapshots from grand canonical Monte Carlo simulations taken under 100 Bar pressure at 77 K (left), 175 K (middle), and 293 K (right) [173].

**Table 1 materials-13-01679-t001:** Comparative study between single-walled carbon nanotubes (SWCNTs) and multi-walled carbon nanotubes (MWCNTs) [2].

SWCNTs	MWCNTs
Single layer of graphene	Multiple layer of graphene
Expensive	Cheaper
Thermal conductivity in the range of 6000 W/m·K	Thermal conductivity in the range of 3000 W/m·K
Semiconducting and metallic properties (excellent field emission capability)	Low physical properties
Bulk synthesis is difficult	Easy to synthesis in bulk
Easily twisted	Difficult to twist
Catalyst needed for synthesis	Manufactured without catalyst
Low purity	High purity
Less accumulation body	Greater accumulation in body
More defection during the functionalization	Less defection, but hard to improve

**Table 2 materials-13-01679-t002:** A comparative study of different types of methods used in the synthesis of CNTs.

Methods	Arc Discharge	Laser Vaporization	Chemical Vapor Deposition	Vapor Phase Growth
Condition	Voltage 25–60 V Current 50–100 A	Temperature 1200 °C and pressure 500 Torr	Temperature 550–1000 °C at atmospheric pressure	Supplying reaction gas and organometallic catalyst in the reactor
Yield	30–90%	~70%	20–100%	-
Carbon Source	Graphite	Graphite	Fossil-based hydrocarbon, botanical hydrocarbon	Hydrocarbon
Advantage	Excellent crystallinity	High quality, high yield compared with arc discharge	Can be controlled	Could be produce in bulk
Disadvantage	Difficult to obtain uniform length nanotube, contain a large amount of impurities	Difficult to maintenance, low production, expensive	Affected the temperature change and position, relatively crystallinity	-
References	[36,37]	[18,23]	[21,22]	[34]

**Table 3 materials-13-01679-t003:** The effects of different carbon nanomaterials (CNMs) on plant and crop growth.

Type of CNMs	Plant	Treatment	Effect	Reference
MWCNTs and oxidized MWCNTs (o-MWCNTs)	*Brassica juncea* (mustard) seeds	23 × 10^−3^ and 46 × 10^−3^ mg/mL of MWCNTs for 5 and 10 days and 2.3 × 10^−3^ and 6.9 × 10^−3^ mg/mL of o-MWCNTs for 5 and 10 days, respectively	After 10 days, seedlings treated with low concentration of o-MWCNTs developed the highest shoot (4.2 cm) and root (5.8 cm) length. Seeds treated with a low concentration of MWCNTs also showed shoot about 1.5 times and root about two times longer than original seeds	[64]
Fullerol and MWCNTs	Tomato seeds	50 mg/L and exposure ranged from 0 to 60 min (0, 5, 10, 30, or 60 min)	When exposed for a short period of 5 min, the germination rate was higher than that of the control group and showed no harm to germination	[65]
Single-walled carbon nanohorns (SWCNHs)	Barley, Corn, Rice, Soybean, Switchgrass, Tomato	25, 50, and 100 μg/mL for 2 and 6 days	The highest germination rate was recorded for barley, corn, rice, and switchgrass seeds exposed to 100 μg/mL SWCNHs and the highest germination rate was observed at 25 μg/mL SWCNHs in tomato seeds	[66]
MWCNTs	Broccoli	10 mg/L MWCNTs, 100 mM NaCl, and 100 mM NaCl + 10 mg/L MWCNTs	The MWCNTs-treated plants had positive effects on growth compared with the control and NaCl alone application	[67]
MWCNTs	Barley, Soybean, Corn	25, 50, and 100 μg/mL for 2 and 6 days	After six days, all seeds treated with MWCNT reached a germination rate of 100% compared with control seeds reaching a germination rate of 63%	[68]
MWCNTs	Tomato plants	50 and 200 μg/mL	The CNT-treated tomato plants produced twice as many flowers as the control plants	[69]
CNTs	Rice	50, 100, and 150 μg/mL	CNTs at appropriate concentrations (~100 μg/mL) promoted rice seed germination and root growth	[70]
SWCNTs and functionalized SWCNTs	Cucumber, Onion	28, 160, 900, and 5000 mg/L for 2 and 3 days	Non-functionalized CNTs enhanced root elongation in onion and cucumber, the effects were more pronounced at 24 h than at 48 h	[71]
SWCNTs (non-functionalized, OH-functionalized, or surfactant stabilized)	Corn	0, 10, and 100 mg/kg	Root length was significantly higher in plants exposed to non-functional SWNT 100 mg/kg and plant root uptake also followed the trend of non-functionalized > surfactant stabilized > OH-functionalized	[74]

**Table 4 materials-13-01679-t004:** The effects of CNMs in the solid-phase extraction (SPE) technique.

CNTs	Analyte	Sample	CNTs Amount (mg)	Recovery (%)	Reference
MWCNTs	Disulfoton sulfoxide, ethoprophos, disulfoton, terbufos sulfone, cadusafos, dimethoate, terbufos, chlorpyrifos-methyl, fenitrothion, malaoxon, pirimiphosmethyl, malathion, chlorpyrifos, disulfoton sulfone, and fensulfothion	Water (run-off, mineral, and tap water)	130	67–107	[94]
GO–MCNTs-diethylenetriamine	Cr(III), Fe(III), Pb(II), and Mn(II) ions	Wastewater	30	95	[95]
MWCNTs	Organophosphate	Garlic	1.2	97–104	[97]
MWCNTs	4-Chlorophenol, 3-chlorophenol, dichlorophenol, trichlorophenol, and pentachlorophenol	River water	300	93–117	[98]
MWCNTs	Tolclofos-methyl, fenitrothin, malathion, phorate, diazinon, isocarbophos, and quinalphos phenamiphos	Peanut oil	100	86–115	[99]
MWCNTs	Ethoprophos, diazinon, fenitrothion, malathion, and phosmet	Agricultural soil, forestal soil, and ornamental soil	100	54–91	[101]

**Table 5 materials-13-01679-t005:** The effects of CNMs in the solid-phase micro-extraction (SPME) technique.

CNTs	Analyte	Sample	CNTs Amount (mg)	Recovery (%)	Reference
MWCNTs	Polybrominated diphenyl ethers (PBDEs)	River water, waste water, milk	20 mg	90–119	[102]
SWCNTs	Ethoprophos, terbufos, thiometon, tefluthrin, iprobenfos, vinclozolin, octachlorodipropyl ether, isofenphos, phenthoate, chlorfenapyr, propiconazol, Ethyl-*p*-nitrophenylthionobenzenephosphonate (EPN), and λ-cyhalothrin	Teas (green tea, oolong tea, white tea, and flower tea)	-	75–118	[103]
SWCNTs	Hexachlorcyclohexan, dichlorodiphenyldichloroethylene, dichlorodiphenyldichloroethane, and dichlorodiphenyltrichloroethane	Lake water	2 g	88–111	[104]
CNTs–silicon dioxide	Diazinon, fenthion, parathion, and chlorpyrifos	River water and agricultural wastewater, pear, grape, and eggplant	50 mg	79–99	[105]

**Table 6 materials-13-01679-t006:** The electrochemical performances of CNTs-based Li-ion batteries. CVD, chemical vapor deposition.

CNMs	Method	Current Density	Initial Discharge Capacity (mA·h/g)	Cycles	Residual Reversible Capacity (mA·h/g)	Reference
CNTs–SnSb_0.5_	CVD	50 mA/g	549	30	369	[111]
CNTs–LiCoO_2_	CVD	0.2 C	118	20	118	[112]
CNTs	arc discharge	2 C	300	300	255	[114]
MWCNTs	arc discharge	0.2 mA cm^−2^	117	30	113	[115]
Short CNTs	CVD	0.8 mA cm^−2^	491	30	170	[119]
Fe_2_O_3_/CNT–graphene foam	CVD	200 mA/g	1190	10	900	[122]
CNTs–cobalt oxide		0.1 C	1250	100	530	[125]
Zn_2_SnO_4_/CNT		100 mA/g	1925.4	30	703.8	[126]

**Table 7 materials-13-01679-t007:** Applications of CNMs in wastewater treatment. COD, chemical oxygen demand.

Applications	Desirable Nanomaterials Properties	Type of CNMs	Efficiency of the CNMs	Reference
Catalysts	Higher catalyst loads and stability, stronger metal–support interactions, high dispersion, high stability and activity, low cost	Ruthenium/MWCNT-COOH-Na_2_CO_3_	98.3% and 70.3% aniline and total organic carbon (TOC) removals	[128]
		Ruthenium/MWCNT-COOH	89.9% and 53.7% aniline and TOC removals	[128]
Mass Transfer	Facilitate contaminant mass transfer, large surface areas, high electrochemical efficiency, degrade organics with much higher current Efficiency and lower energy consumption	CNTs	The efficiency was 340–519% higher than the conventional reactor, and the energy consumption was only 16.5–22.3% of the conventional reactor	[131]
Adsorption	Large specific surface areas, high chemical and thermal stabilities, high aspect ratios, exceptional mechanical strength, diverse contaminant–CNT interactions	SWCNTs, MWCNTs	The maximum zinc adsorption capacities of SWCNTs and MWCNTs were 43.66 and 32.68 mg/g, respectively, in the initial zinc ion concentration range (10–80 mg/L)	[134]
Flocculation	Exceptional adsorption capabilities and efficiencies, larger surface area, affinity towards target compounds	CNTs	Demonstrated the ability to successfully coagulate colloidal particles in the brewery wastewater	[135]
Electrode	Effective compound adsorption and oxidation, high energy efficiency, fast reaction rate, electrochemical oxidation	Ti/SnO_2_-Sb-CNT electrode	80.12% and 46.01% COD and TOC removals	[136]

**Table 8 materials-13-01679-t008:** The effects of the carbon-based electrode on microbial fuel cells (MFCs).

Type of Electrode		MFCs	Type of MFCs	Effect	Power Density (Max.) mW/m	Reference
Anode	Cathode					
Graphite fiber	Carbon nanotube/Pt	Effluent from an air-cathode MFC	Single chamber MFCs	The cathode had a maximum power density of about two times higher than that of the carbon cloth cathode	329	[144]
Carbon paper	CNTs/Poly-pyrrole	Anaerobic digester sludge collected from Indah Water Konsortium treated Palm oil mill effluent (POME)	Two cubic shaped chambers	COD removal of the system using CNT/PPy was 96%	113.5	[145]
Carbon cloth	N-CNTs on carbon cloth	Acetate-laden synthetic wastewater	Air-cathode cylindrical-shaped MFCs, dual chamber	The maximum power density was about 9% higher than that of Pt-carbon on carbon cloth	135	[147]
Carbon paper	CNTs/Pt	Palm oil mill effluent (POME- Selangor, Malaysia) sludge	Two cylindrical H-shaped chambers	The composite electrode increased the power output of MFC by 8.7~32% compared with Pt electrode	169.7	[149]
Carbon paper	Chemically activated carbon nanofibers	Palm oil mill effluent (POME) anaerobic (Selangor, Malaysia) sludge	Two cylindrical and H-shaped chambers	COD removal was approximately 82.3% and could generate up to 3.17 times more power than carbon paper	61.3	[150]
Carbon fiber	Nitrogen-doped CNTs (N-CNTs)	20% domestic wastewater collected from a municipal wastewater treatment plant of Shanghai, China	Air-cathode single chamber MFCs	The power density drop rate was low, so electricity can be produced more permanently than the platinum catalyst	1600 ± 50	[151]
MWCNT/ rGO-biofilm	carbon fiber brush	*S. putrefaciens* CN32 cell suspension was inoculated on bacteria	H-type dual-chamber	Composite electrodes provide higher maximum power density than individual MWCNTs and rGO	789	[152]
Vertically Aligned CNTs	Cr/Au film	Acetate-fed microbial electrolytic cells (MEC) with Geobacter-enriched bacterial community from anaerobic digestion sludge Dual chamber MFCs; anode and cathode chambers		61.3% of Coulombic efficiency	270	[153]
Randomly Aligned CNTs	Cr/Au film			73% of Coulombic efficiency	540	
Spin/spray layer-by-layer CNTs	Cr/Au film			73% of Coulombic efficiency	540	
CNTs/polyaniline (PANI)	Pt	Bacteria	*E. coli*-based MFCs	Composite electrodes containing 20 wt.% CNTs provide high discharge performance and high power output	42	[154]
CNTs	CNTs/Pt	Bacteria	Air-cathode MFCs	COD removal was 95% and the maximum coulombic efficiency was 67%	65	[155]
rGO-CNT sponges	-	Anaerobic sludge	Aerobic chamber and anoxic chamber	Produced higher durability	Max. current density of 335 A m^−3^	[156]

**Table 9 materials-13-01679-t009:** Hydrogen storage efficiency of different kinds of CNMs at different conditions.

CNMs	Storage (wt.%)	Temperature (K)	Pressure (MPa)	Reference
CNTs	9.6	77	10	[159]
CNTs	1.5	296	12.5	[160]
SWCNTs	4.5	77	6	[161]
Chemically activated carbon	5.6	77	4	[162]
Carbon with boron	5.9	298	10	[163]
SWCNTs	1.73	77	10	[164]
SWCNTs-SnO_2_	2.4	623	5	[165]
Si-doped SWCNTs	2.5	298	10	[166]
Un-doped SWCNTs	1.4	298	10	

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
