# Peer review of "Carbon Nanotubes-Based Nanomaterials and Their Agricultural and Biotechnological Applications"

_materials, 2020, doi:10.3390/ma13071679_

Round 1

Reviewer 1 Report

The paper reports a concise review of carbon nanotubes-based nanomaterials for agricultural and biotechnological applications.

The Authors have made a great review of the literature and have discussed the topic in a very interesting way. The paper is written well, but I recommend expanding the summary and improve the quality of the figures.

I think that the manuscript has sufficient scientific quality and relevance for Materials. I suggest accepting the publication as it is after adding better quality figures and expanding the summary.

Author Response

Point 1: The paper reports a concise review of carbon nanotubes-based nanomaterials for agricultural and biotechnological applications. The Authors have made a great review of the literature and have discussed the topic in a very interesting way. The paper is written well, but I recommend expanding the summary and improve the quality of the figures. I think that the manuscript has sufficient scientific quality and relevance for Materials. I suggest accepting the publication as it is after adding better quality figures and expanding the summary.

Response 1: The paper reports a concise review of carbon nanotubes-based nanomaterials for agricultural and biotechnological applications. The Authors have made a great review of the literature and have discussed the topic in a very interesting way. The paper is written well, but I recommend expanding the summary and improve the quality of the figures. I think that the manuscript has sufficient scientific quality and relevance for Materials. I suggest accepting the publication as it is after adding better quality figures and expanding the summary.

Reviewer 2 Report

In this work authors summarize the present knowledge about carbon-based nanomaterials and they focus on their agricultural and biotechnological applications. The following question must be considered prior publication:

  1. Authors define their work in the abstract both as a review article and as a feature article. These two types are different works, chose one of them and improve the work based on the criteria of the chosen type.
  2. The title is about carbon-based nanomaterials, while the work is about only carbon nanotubes. The title must be revised based on the topic of the manuscript.
  3. The well-known general information about carbon nanotubes from the introduction must be deleted.
  4. In many cases the figures and tables are incorrect. E.g. Fig.1. length of MWCNT missing, data are not general, the caption is incomprehensible. Tab.1. data are not up-to-date, MWCNT’s line 5 and 6 are the same.
  5. Although the manuscript summarizes 173 references, in some cases the references are missing. E.g. line 60.
  6. In a review article the authors must cite the original works instead of other review articles. Ref. 11, 12, 13, 18 etc.
  7. More than half of the references are before 2009, less than 20% after 2016. From 2018, I only found 4 references, and these are the latest. The results of the last five years must also be presented.

In general, despite the title (the agricultural and biotechnological applications), the described topic is too broad and general (synthesis, general properties), rather than a critical analysis of the present literature. Section 1-3. should be rewritten and shortened based on the aim of the work and Section 4. should be extended based on the present knowledge and literature (2016- today).

It might be reconsidered after a major revision providing manuscript consistent with presently available literature.

Author Response

In this work, authors summarize the present knowledge about carbon-based nanomaterials and they focus on their agricultural and biotechnological applications. The following question must be considered prior publication:

Reply to Reviewer: We appreciate the reviewer positive recommendations for the manuscript and their valuable comments for the future improving the paper.

Point 1: Authors define their work in the abstract both as a review article and as a feature article. These two types are different works, chose one of them and improve the work based on the criteria of the chosen type.

Response 1: Thank you very much for reviewing the manuscript. We agree with the point of view. This is a review article, and based on this we have improved the quality of the manuscript.

Point 2: The title is about carbon-based nanomaterials, while the work is about only carbon nanotubes. The title must be revised based on the topic of the manuscript.

Response 2: Thank you very much for reviewing the manuscript. Yes, this review highlighted the applications of carbon nanotubes-based nanomaterials for agricultural and biotechnological applications. We have revised the title of the manuscript as per your suggestion.

Point 3. The well-known general information about carbon nanotubes from the introduction must be deleted.

Response 2: Thank you very much for your kind suggestion. We have removed the well-known general information about carbon nanotubes from the introduction section of the manuscript.

Point 4. In many cases the figures and tables are incorrect. E.g. Fig.1. length of MWCNT missing, data are not general, the caption is incomprehensible. Tab.1. data are not up-to-date, MWCNT’s line 5 and 6 are the same.

Response 4: Thank you very much for reviewing the manuscript. We have resolved the mentioned demerits from the revised manuscript as per your suggestion.

Point 5: Although the manuscript summarizes 173 references, in some cases the references are missing. E.g. line 60.

Response 5: Thank you very much for reviewing the manuscript. We have added the additional references in the revised manuscript.

Point 6. In a review article, the authors must cite the original works instead of other review articles. Ref. 11, 12, 13, 18 etc.

Response 6: Thank you very much for your kind suggestion. We have also cited the original work in the revised manuscript with other articles.

Point 7: More than half of the references are before 2009, less than 20% after 2016. From 2018, I only found 4 references, and these are the latest. The results of the last five years must also be presented.

Response 7: Thank you very much for your kind suggestion. We have added the results of the last five years in the revised manuscript with proper citation.

In general, despite the title (the agricultural and biotechnological applications), the described topic is too broad and general (synthesis, general properties), rather than a critical analysis of the present literature. Section 1-3. should be rewritten and shortened based on the aim of the work and Section 4. should be extended based on the present knowledge and literature (2016- today).

It might be reconsidered after a major revision providing manuscript consistent with presently available literature.

Response: Thank you very much for reviewing the manuscript. We have modified the manuscript section as per your kind suggestion

Reviewer 3 Report

Recommendation: Publish after major revisions noted.  

Comments:  

This manuscript summarized the applications of carbon nanotubes-based nanomaterials mainly for agriculture. The authors should consider major revisions. 

  1. This manuscript is in need of substantial editing and English improvement.
  2. Please consider to modify the title. Current manuscript is not a concise review. 
  3. Page 3, line 60: The authors only focused on CNTs not other carbon-derived nanomaterials. Please consider to revise. 
  4. Please consider to expand the biotechnological field. 
  5. Consider to elaborate toxicity and pathology caused by CNTs. Mention the factors affecting the toxicity of CNTs. 
  6. Page 9, line 265: “Each sensor has a different working principle and significant”. Consider to modify the sentence and explain briefly. 
  7. Consider to include critical advances toward commercial applications.

Author Response

Recommendation: Publish after major revisions noted.

Reply to Reviewer: We appreciate the reviewer positive recommendations for the manuscript and their valuable comments for the future improving the paper.

Comments: This manuscript summarized the applications of carbon nanotubes-based nanomaterials mainly for agriculture. The authors should consider major revisions. 

Point 1: This manuscript is in need of substantial editing and English improvement.

Response 1: Thank you very much for reviewing the manuscript. We have improved the English quality of the revised manuscript.

Point 2: Please consider modifying the title. The current manuscript is not a concise review. 

Response 2: Thank you very much for your kind suggestion. We have modified the title of the manuscript.

Point 3: Page 3, line 60: The authors only focused on CNTs, not other carbon-derived nanomaterials. Please consider to revise. 

Response 3: Thank you very much for your kind suggestion. We have modified the sentence in the revised manuscript.

Point 4: Please consider to expand the biotechnological field. 

Response 4: Thank you very much for your kind recommendation. We have expanded the biotechnological applications of CNTs-based nanomaterials for wastewater treatment in the revised manuscript.

Point 5: Consider to elaborate toxicity and pathology caused by CNTs. Mention the factors affecting the toxicity of CNTs. 

Response 5: Thank you very much for reviewing the manuscript. We have mentioned the factors affecting the toxicity of CNTs for seed germination or plant growth in the revised manuscript with proper citation.

Various factors such as size, shape, surface structure, solubility, concentrations, as well as the presence of the functional groups have significant contributions towards the toxicity and pathology caused by CNTs in the germination of seeds.61, 72

Point 6: Page 9, line 265: “Each sensor has a different working principle and significant”. Consider to modify the sentence and explain briefly. 

Response 6: Thank you very much for reviewing the manuscript. We have modified the sentence with a brief explanation in the revised manuscript.

Different types of sensors are explored for monitoring the pollutant/species present in the medium. Biosensors are utilized to detect compounds such as aromatic and organic compounds, and halogenated pesticides. Solid-state electrochemical sensors are suitable for the chemical gas sensors from their sensitivity, reproducibility, and power consumption. The basic principle of a biosensor for soil diagnosis is to determine the relative activity of favorable and unfavorable microbe’s presence in the soil based on differential oxygen consumption due to the respiration. Surface Plasmon resonance (SPR) phenomenon is also explored for the development of the biosensor from metallic nanoparticles.79

Point 7: Consider to include critical advances toward commercial applications.

Response 7: Thank you very much for reviewing the manuscript. We have added the commercial applications of CNTs or their derivatives in the field of wastewater treatment.

It has been noted that phenolic compounds are often explored in the commercial manufacturing of several products such as resins, polymeric materials, ion exchange resin, dyes, drugs, explosives, and other products. Due to the extensive uses of phenolic products, a large amount of phenol is discharging from industries in the water, which causes the toxicity and can damage the cellular proteins. Therefore, the removal of phenolic compounds from the contaminated water on large scale is necessary for a healthy life. For this, CNTs with rich pore structure, chemical abilities, high surface area, and sharp curvatures show the great potential for the removal of the phenolic compounds from the contaminated water through π-π, electrostatic, hydrophobic, and hydrogen bonding interactions.137

Round 2

Reviewer 2 Report

It is very difficult to review the literature that has been reported until the present day and publish an up-to-date summary of a current topic because day by day more and more works are available. I still believe that more results from the last five years should be presented. The authors made efforts to improve the quality of the manuscript and answered several questions, therefore, I suggest publishing the work in Materials.

Reviewer 3 Report

Comments: 

I reviewed the authors' response to the critiques of the original submission and noticed that the manuscript is revised adequately. 

Recommendation: Publish